# Modulation of Inflammasome and Pyroptosis by Olaparib, a PARP-1 Inhibitor, in the R6/2 Mouse Model of Huntington’s Disease

**DOI:** 10.3390/cells9102286

**Published:** 2020-10-13

**Authors:** Emanuela Paldino, Vincenza D’Angelo, Daunia Laurenti, Cecilia Angeloni, Giuseppe Sancesario, Francesca R. Fusco

**Affiliations:** 1Laboratory of Neuroanatomy, IRCCS Fondazione Santa Lucia, 00143 Rome, Italy; e.paldino@hsantalucia.it (E.P.); daunia25@gmail.com (D.L.); cecilia.angeloni@live.it (C.A.); 2Dipartimento di Medicina dei Sistemi, Università di Roma 2 Tor Vergata, 00133 Roma, Italy; dangelo@med.uniroma2.it (V.D.); sancesario@med.uniroma2.it (G.S.)

**Keywords:** Huntington’s disease, Parp-1 inhibition, Inflammasome, Pyroptosis, NLRP3, caspase-1, microglia

## Abstract

Pyroptosis is a type of cell death that is caspase-1 (Casp-1) dependent, which leads to a rapid cell lysis, and it is linked to the inflammasome. We recently showed that pyroptotic cell death occurs in Huntington’s disease (HD). Moreover, we previously described the beneficial effects of a PARP-1 inhibitor in HD. In this study, we investigated the neuroprotective effect of Olaparib, an inhibitor of PARP-1, in the mouse model of Huntington’s disease. R6/2 mice were administered Olaparib or vehicle from pre-symptomatic to late stages. Behavioral studies were performed to investigate clinical effects of the compound. Immunohistochemical and Western blotting studies were performed to evaluate neuroprotection and the impact of the compound on the pathway of neuronal death in the HD mice. Our results indicate that Olaparib administration starting from the pre-symptomatic stage of the neurodegenerative disease increased survival, ameliorated the neurological deficits, and improved clinical outcomes in neurobehavioral tests mainly by modulating the inflammasome activation. These results suggest that Olaparib, a commercially available drug already in use as an anti-neoplastic compound, exerts a neuroprotective effect and could be a useful pharmaceutical agent for Huntington’s disease therapy.

## 1. Introduction

Huntington’s disease (HD) mutation consists of a CAG expansion beyond the normal 10–35 repeats of the IT15 gene [1,2,3] localized on the short arm of chromosome 6, encoding the huntingtin protein [4]. HD pathology is characterized by an extensive degeneration of the striatal part of the basal ganglia, where medium spiny neurons progressively die along with parvalbumin interneurons, while cholinergic and somatostatinergic interneurons survive [5]. Microscopically, neuronal intranuclear inclusions constituted by mutated huntingtin [6] are observed in most cells, even those that are not involved in the degeneration process [7]. Such inclusions interact with and impair several cellular functions [8]. There is debate about the involvement of neuronal intranuclear inclusions (NIIs) in neuronal death. In particular, intranuclear htt-aggregates were speculated to be involved with apoptosis [9]. The role of apoptosis has been widely discussed in HD [10,11], and several studies revealed the presence of apoptotic cells and DNA degradation products both in HD patients’ brains [11,12,13,14] and in experimental models of HD [11,15,16]. An emerging role for other forms of neuronal death has more recently gained momentum.

Pyroptosis is a type of cell death that is caspase-1 (Casp-1) dependent and leads to rapid cell lysis [17]. Similar to apoptosis, pyroptosis is a programmed cell death, but it depends on a different caspase. Indeed, several features of pyroptosis are also seen in apoptosis, but, when examined more carefully, the two processes are fairly distinct. For example, pyroptosis involves DNA damage, and TUNEL assays are positive [17]. In apoptosis the nucleus becomes pyknotic and subsequently breaks up in karyorrhexis, while the nucleus in pyroptosis undergoes chromatin condensation, but remains intact. Moreover, no karyorrhexis occurs in pyroptosis. Interestingly, pyroptosis is associated with neuroinflammation. Research on the involvement of inflammasome in neurodegeneration is gaining momentum in several CNS disorders [18].

Inflammasomes are multiprotein oligomers that are assembled in response to several cellular stresses, and their activation leads in turn to the recruitment and activation of procaspase-1 [19] The major components of the inflammasome complex are pattern recognition receptors (PRRs) [20]. A number of inflammasomes have been described, which include the NLR family pyrin domain containing 3 (NLRP3), NLRP1, AIM2, and others. The NLRP3 inflammasome was found to be associated with inflammatory and immune system-related disorders [21]. Furthermore, the NLRP3 inflammasome was discovered to be involved with the pathogenesis of a several neurodegenerative diseases [22]. We have recently shown [23] the involvement of pyroptosis in association with the NLRP3 inflammasome in HD neurodegeneration in a mouse model of HD. Unrestrained activation of Poly (ADP-ribose) polymerases (PARP), a group of enzymes involved in regulation of several cell functions, is induced by DNA damage and is able to deplete intracellular NAD stores leading to cell death [24,25]. Indeed, an overactivation of PARP has been associated with the pathogenesis of several nervous system disorders, such as excitotoxicity, traumatic injury, ischemia-reperfusion, and inflammation [26,27,28,29,30]. It has been shown that an unbalanced activation of PARP is able to evoke the release of apoptosis-inducing factor (AIF) [31,32]. Excessive PARP activation also leads to the translocation of AIF from the mitochondria to the nucleus, thereby causing a programmed cell death named parthanatos [33], which is a mechanism that is not associated with caspase. It has been shown that inhibition of PARP by specific compounds can prevent the release of AIF to the nucleus, and therefore promotes cell survival [33,34]. In a previous study, our group showed the beneficial effects of a PARP inhibitor in the mouse R6/2 model of HD [35]. In that study, we associated the beneficial effects of PARP inhibition with an anti-apoptotic reaction, measured by the expression of the markers of apoptosis Bcl-2 and Bax in INO-1001 treated R6/2 mice [35]. In the present study, we used another PARP-1 inhibitor, namely, Olaparib, a drug already in use for patients with ovarian cancer, in the transgenic model of Huntington’s disease.

We aimed at clarifying two main aspects: first, the effects of PARP inhibition by a different, commercially available drug on neuroprotection in HD, and second, to evaluate the impact of this compound on apoptotic and pyroptotic cell death in the striatum.

## 2. Materials and Methods

### 2.1. Animals and Drug Administration

Animals satisfying ARRIVE guidelines were used to perform experiments in accordance with the European Communities Council Directive (2010/63 EU), as adopted by the Santa Lucia Foundation of Animal Care and Use, and approved by the Italian Ministry of Health. Transgenic female R6/2 mice carrying the mutant human HTT exon 1, which determines the abnormal expanded CAG repeats, were kept in coupling with B6CBAF1/J males (Jackson Laboratories Bar Harbor, ME). All experiments were performed using F1 mice. Wild type and R6/2 mice (20 mice/per experimental group) were treated intraperitoneally with either vehicle (0.9% saline) or Olaparib dissolved in saline (10 mg/kg/day) every day until sacrifice. Treatment started at 4 week of age and ended at 13 weeks of age. Mice were identified by a randomly assigned code and housed four per cage under conventional laboratory conditions (room temperature: 20 ± 2°C; humidity: 60%) and a 12/12 h light/dark cycle (7:00 a.m.–7:00 p.m.) with ad libitum access to food and water. Observers who were blinded to genotype and treatment collected all the experimental data.

### 2.2. Survival and Weight

The survival study, according to Hersh and Ferrante [36] was conducted following the criterion for euthanasia, which is the point when animals were not able to right themselves after 30 s when placed on their side. Experimental mice were weighed twice a week starting from the beginning of treatment until sacrifice. Their weights were recorded, and every weight variation was plotted.

### 2.3. Animals Behavior 

**1.** Clasping R6/2 mice exhibit a specific hind-limb clasping phenotype when suspended by the tail. The clasping phenotype has been extensively studied and used to recognize the neurological impairment in HD mice, and it is considered a measure of disease progression. Mice were suspended by their tail for 60 s, and the total time spent clasping the hind limb was recorded twice weekly.

**2.** Rotarod. The five-station Rotarod performance test (Rotarod/RS LSI Letica, Biological Instruments, Varese, Italy) was used to estimate mouse motor coordination and balance. Mice performed the Rotarod test twice weekly from 4 to 13 weeks of age. Three-trial measurements on the rod for their latency or fall were recorded.

### 2.4. Tissue Processing and Immunohistochemical Studies

Eighty animals (20 R6/2 treated with vehicle, 20 R6/2 treated with Olaparib, 20 vehicle and 20 Olaparib-treated Wild type mice) were transcardially perfused under deep anesthesia with saline solution containing 0.01 mL heparin, brains were removed and cut in half. Tissue sectioning was performed on a sliding frozen microtome at 40 μm thickness. All brain sections were processed for single label EM-48 ubiquitin (1:500, Chemicon, Temecula, CA); NLRP3 (1:200, Abcam, Novus Biologicals, Italy); Calbindin (1:500, Abcam, Novus Biologicals, Italy); Parvalbumin (1:200, Chemicon International, Inc., Temecula, CA, USA); Calretinin (1:200, Chemicon International, Inc., Temecula, CA, USA), and GFAP (1:600 polyclonal anti-GFAP, Millipore, Italy). We used Alexa Fluor 488 and Alexa Fluor 555 (Immunological Sciences, Italy) as secondary antibodies. Brain sections from the same level of the bregma, were mounted on gelatin-coated slices and cover slipped with Gel-Mount. Samples were examined with the support of confocal laser scanner microscopy (Zeiss LSM 800); images were acquired and subsequently analyzed to quantify the immunofluorescence intensity of Calbindin, NLRP3, Parvalbumin, Calretinin, and GFAP-positive cells. To evaluate pyroptosis, peroxidase-antiperoxidase diaminobenzidine tetrahydrochloride single-label immunohistochemistry for caspase-1 was performed [23].

### 2.5. Western Blotting

Dissected striata from saline or Olaparib-Wt and R6/2 half brains were homogenized with the RIPA lysis buffer containing a protease and phosphatase inhibitor cocktail (Sigma Aldrich, USA) and centrifuged at 13,000 × g for 20 min. Equal amounts of protein were separated using sodium dodecyl sulfate-polyacrylamide gel electrophoresis, transferred to polyvinylidene fluoride membranes, and incubated with rabbit NLRP3, (1:1000 Abcam, Novus Biologicals), mouse caspase-1 (1:1000, Abcam, Novus Biologicals, Italy), rabbit pAKT, AKT (1:1000 Cell Signaling Technology), rabbit pCREB, CREB (1:1000 Immunological Sciences, Italy), rabbit pERK, ERK (1:1000 Immunological Sciences, Italy), mouse PSD95 (1:1000, Abcam Novus Biologicals, Italy), rabbit caspase-8 (1:1000, Abcam, Novus Biologicals, Italy), rabbit PARP-1 (1:1000, Abcam, Italy), and mouse GAPDH (1:10,000; Sigma Aldrich, St Louis, MO, USA) antibodies, overnight at 4 °C. After being washed with Tris-buffer saline (TBS)/Tween 20, membranes were incubated with HRP-labeled secondary antibodies. Protein signals were visualized using the Invitrogen iBright CL 1500 Imaging system.

### 2.6. Microglia Morphological Characterization

Microglial morphology was studied by immunostaining brain sections with an antibody labeling microglia (1:500 goat anti-Iba-1 from Abcam Novus Biologicals, Italy). Striatal brain sections were incubated with the primary antibody for 72 h at 4 °C, followed by incubation with the secondary antibody for 2 h at room temperature. Images were acquired with confocal laser scanner microscopy (Zeiss LSM 800) in order to perform fractal analysis. Microglia cells in the area of interest were captured using 20× magnification with a 2× zoom objective, producing images in the format of 1024 × 1024, and Airy Units 1.0. This configuration was used for all samples. The protective or toxic phenotype was characterized by using the free software Frac Lac for Image J available at the Fiji Image J website. Five microglia were randomly chosen for fractal analysis within each photomicrograph (4 photomicrographs per animals) for a total of 200 cells analyzed per animal group. Binary images were converted to outlines using Image J and the performed analysis included Iba-1 positive cells’ fractal dimension, lacunarity, density, area, and perimeter.

### 2.7. Statistical Analysis 

The data collected were analyzed to compare the effect of Olaparib on weight, clasping, rotarod, NIIs percentage and size, and CALB, NLRP3, PARV, CALR, GFAP, Iba-1-expression in the striatum of differently treated mouse groups. Statistical analysis was performed by ANOVA available on the software GraphPad Prism version 8.0. The *p* values < 0.05 were considered statistically significant. All confocal images were acquired under no saturation conditions, with a 20× and 63× objective raising a 1× or 2× zoom, respectively, with a value 0 of Offset, producing images in the format 1024 × 1024, Airy Units 1.0. Cells of interest were selected using the freehand tool. From the Analyze menu, Set measurements Mean “Grey Value”, “Area”, and “Min and Max Grey Value” were selected. The region characterized by absence of fluorescence was considered in the background and it was subtracted. Finally, the mean values with SEM were obtained for all measures. ANOVA analysis available in the software GraphPad Prism version 8.0 was performed. The *p* values of less than 0.05 were considered statistically significant. The same set of configurations was performed for all samples. The colocalization Pearson’s coefficients of NLRP3 in striatal projection neurons were calculated by performing JacoP analysis using the plugin provided by Fiji ImageJ.

## 3. Results

### 3.1. R6/2 Survival and Weight Changes

Olaparib treatment proved to have a beneficial effect on R6/2 mice. As shown by the Kaplan–Meyer survival curve, Olaparib treated animals displayed a statistically significant increase in survival, compared to the saline treated group (Figure 1A).

A graph describing all weight changes among the different groups is shown in Figure 1B. At 13 weeks, R6/2 mice treated with Olaparib weighed 22 ± 1.45, whereas vehicle treated R6/2 mice weighed 16 ± 0.45g, thus showing a beneficial effect of the drug over time (Genotype X Treatment X Time F_8,250_=12.78 *p* < 0.000).

### 3.2. Olaparib Improves Neurological Deficits in R6/2 Mice

#### 3.2.1. Clasping

Olaparib daily administration mitigated the progression of neurological abnormalities in the R6/2 mice. No clasping activity was present in wild type mice. The time spent clasping was significantly less, and the phenotype appeared later in the Olaparib treated group than in the vehicle group, as shown in Figure 1C (*p* < 0.001).

#### 3.2.2. Motor Behavior

The Rotarod apparatus evaluated motor behavior performances of mice. R6/2 mice had a statistically significant impairment in motor coordination compared to wild type mice F_1,36_=59.39 *p* < 0.0001; the three-way ANOVA showed a significant improvement of motor performance F_1,36_ = 21.35 *p* < 0.0001 after Olaparib treatment in R6/2 mice (Figure 1D).

### 3.3. Evaluation of NIIs Number and Size

The expression of the exon1 of mutant huntingtin in the R6/2 mice results in the formation of neuronal intranuclear inclusions (NIIs) detected with the antibody EM-48. The analysis of EM-48 immunofluorescence in the striatal brain region of 13-week-old R6/2 mice treated with Olaparib, counterstained with Neurotrace^™^, showed that the number of NIIs was not significantly reduced compared to the vehicle-treated group (Figure 2G). However, the analysis of all Olaparib-treated R6/2 mice revealed a statistically significant reduction of the size of EM-48 immunoreactive NIIs (Figure 2H). The beneficial effect of Olaparib on R6/2 mice was also assessed by analyzing the expression levels of specific markers of the HD neurodegenerative process. Indeed, Olaparib-treated R6/2 mice showed the restoration of the CREB, ERK, and AKT activation necessary for survival of striatal neurons (Figure 3A,B). Moreover, the postsynaptic activity, as identified by PSD-95 protein, was up-regulated in R6/2 treated with Olaparib compared to that of vehicle-treated R6/2 mice.

### 3.4. NLRP3 Expression in Striatal Projection Neurons

Olaparib treatment did not induce the 100% inhibition of PARP protein expression levels in the R6/2 mice (Appendix A), as was previously described for all PARP inhibitors [37]. However, we investigated the effect of this inhibitor on R6/2 mice inflammasome. We investigated the distribution of the pyroptosis marker NLRP3 [23] in the striatal projection neurons, labeled by Calbindin immunoreactivity. Calbindin labeled striatal projection cells were observed in the 13-weeks-old Wt mice, where they constituted the majority of neurons, whereas they were dramatically reduced in the R6/2 mice, where we observed a statistically significant enhancement in NLRP3 immunofluorescence intensity (Figure 4 G–J). Olaparib treatment rescued Calbindin positive striatal projection neurons and promoted a statistically significant down-modulation of NLRP3 (Figure 4K–M). In the Olaparib-treated R6/2 mice, in which we observed the down-modulation of NLRP3, the cleaved product of caspase-1 was significantly down-modulated, showing an effect on the pyroptosis activation pathway (Figure 4R–S).

### 3.5. Analysis of NLRP3 Expression in the Striatal Interneurons 

Along with the distribution of NLRP3 in projection neurons, we studied the pattern of its localization in the different interneuron subtypes, each of which expresses a peculiar susceptibility to HD degeneration. The intensity of NLRP3 immunoreaction products in the interneuron subtypes was investigated in all experimental groups (Figure 5A,B). We confirmed a significant reduction of CALR and PARV interneurons in the vehicle-treated R6/2 mice, where NLRP3 expression, in terms of intensity of immunoreactivity, was significantly increased. NLRP3 expression levels culminated in the vehicle treated R6/2 mice in which the number and the immunofluorescence intensity of Parvalbumin-containing neurons were significantly decreased. The mean calculated Pearson’s colocalization coefficients were near 1 (Figure 5 D–F). Olaparib treatment was able to promote the survival of Calretininergic and Parvalbuminergic interneurons.

### 3.6. Microglial and Astrocytic Activation in R6/2 Mice

Olaparib treatment promoted a statistically significant reduction of cells positive for GFAP in R62 mice (Figure 6 A–D). Iba-1 immunofluorescence was performed to study the activation of microglia in the different experimental groups. Both vehicle and Olaparib-treated wild type mice displayed a ramified or primed microglia morphology, with a bigger cell body, but with processes similar to that of ramified microglia. The presence of dystrophic microglia in the 13-week-old wild type mice can be attributed to the aging/senescence process. The immunostaining for Iba-1 in vehicle-treated R6/2 mice revealed an intense microglial reaction, where microglial cells appeared in large quantities and displayed an amoeboid cell body (Figure 6G). Fractal analysis revealed a different pattern of microglia ramification in the R6/2 treated with Olaparib compared to that of vehicle. Frac Lac for Image J calculated fractal dimension, lacunarity, density, area, and perimeter that are measures of microglia complexity. In the Olaparib-treated R6/2 mice we observed an increment of fractal dimension, area, and perimeter with a simultaneous decrease of lacunarity and density measures, revealing the presence of a primed or ramified microglia (Figure 6J–M).

## 4. Discussion

PARP-1 is activated by events that lead to DNA damage, and it is implicated in several physiological functions such as modulation of inflammatory pathways, transcription, and chromatin remodeling [38].

Olaparib is an FDA-approved PARP inhibitor that is already available and commonly used to treat ovarian cancer. Indeed, the role of PARP in DNA repair provided the rationale for the use of this compound, based on the hypothesis that by inhibiting PARP, a suppression of the repair of chemotherapy and radiotherapy-related DNA damage would occur, thereby allowing the death of cancer cell [39,40].

Beneficial effects of Olaparib were also hypothesized in other non-oncological diseases, particularly conditions where DNA damage constitutes part of the pathophysiological response [41]. Indeed, it was demonstrated that, in neurons, PARP activation is not only a downstream player of NMDA-receptor-activation-induced neurotoxicity, but also enhances the expression of calcium channels that are responsible for a delayed type of neuronal death [42]. When PARP is inhibited, the viability of cells that are subjected to oxidative stress can be ameliorated. In the context of neuroinjury, a therapeutic opportunity exists not only to reduce the delayed loss of neurons associated with brain ischemia (stroke, cardiac arrest, or trauma) but also to decrease the late dementia that frequently occurs within a few months of brain ischemic events.

One of the aims of this study was to evaluate the effects of PARP-1 inhibition by Olaparib in the R6/2 mouse model of HD. A previous study by our group [35] had led to encouraging results, stimulating further studies on PARP-1 inhibition and unveiling the mechanisms of neuroprotection in an HD model. In that study, we described a positive effect on neuroprotection that was associated to a reduction in the number of NIIs, on one hand, and to reduced apoptosis, on the other. INO-1001 was able to increase CREB and BDNF activation in medium spiny neurons and to dramatically reduce cell death in the striatum. In the present study, we utilized a PARP-1 inhibitor that is already in use for other diseases to possibly accelerate the process towards offering a viable pharmaceutical compound to HD patients, and we describe how Olaparib was beneficial in the primary outcome measures. The clinical manifestation of the disease, that in the R6/2 model we identify as the clasping behavior, was milder in the treated group than in the control group, and appeared later in the course of the disease. Moreover, Olaparib-treated animals survived longer than the vehicle treated ones. In addition, from a clinical standpoint, we observed that Olaparib improved significantly the rotarod performances. Thus, we drew the inference that Olaparib is beneficial for HD symptoms.

In our immunohistochemical studies, we describe a reduction in the size of NIIs, rather than a reduction in their numbers. NIIs are aggregates of mutated huntingtin that can sequester several proteins [43]. We speculate that this is an epiphenomenon of a reduced sequestration of those survival factors that are down-modulated in HD pathology.

Sparing of striatal projection neurons, and also of selected subpopulations of interneurons, was observed as an effect of neuroprotection exerted by Olaparib. In the surviving neurons, the protein expression levels of pCREB, pERK, and pAKT were increased compared to that of the saline group.

By studying the effects of PARP-1 inhibition, we focused our attention on the inflammasome-pyroptosis processes, which are known to be implicated in HD neuropathology [23]. We describe that Olaparib was able to reduce the expression of NLRP3 with subsequent inactivation of caspase-1, on one hand, and a very specific modulation of number and morphology of microglial cells, on the other. Therefore, as per the second aim of the study, we confirmed the postulated effect on pyroptosis in the HD model.

Inflammation is involved in the pathogenesis of many neurological disorders [44,45]. The major cell type responsible for neuroinflammation in the brain is the microglia. Cytokines, nitric oxide, and reactive oxygen species are mediators of the neuropathology deriving from microglial cells [46]. Microglial activity is highly involved with the inflammasome in HD [23].

Inflammasomes are complexes of proteins assembled in response to various cellular stresses, and their activation leads in turn to the recruitment and activation of procaspase-1. NLRP3 inflammasome displays a major role in inflammatory and immune system-related disorders and its involvement in HD has been clearly demonstrated [47,48].

The effects of Olaparib on microglial cells and on NLRP3 are, therefore, a further proof of the interplay between inflammation and neurodegeneration in HD.

When examining the effects of PARP inhibition, in general, and of Olaparib, in particular, one might ask how it is possible that a therapy can be beneficial as a cytostatic (in oncology) and also as a promoter of cell survival (in neurodegenerative diseases such as HD). A very important aspect that determines whether PARP inhibition has a positive or negative effect on cell viability is time. In general, PARP inhibitors are protective for cells that are in non-replicating states or that are slowly replicating and so have enough time to repair DNA before the defects have detrimental effects. Conversely, in cells that replicate rapidly, such as neoplastic cells, the effects of PARP inhibitors on cell survival is detrimental [37].

We show that Olaparib also upregulated, to some extent, PARP expression in the wild type mouse. This should not be surprising, as a genotoxic effect of PARP inhibitors has been widely described [37]. However, the beneficial effects of these compounds greatly exceed the detrimental ones.

PARP is not only involved in specific forms of cell death, but is also implicated in signal transduction events and in the promotion of several pro-inflammatory signaling pathways. Thus, it can be asserted that the events of cell necrosis and the processes of inflammation are intertwined and can lead to a positive feedforward cycle. These events promote more pathophysiological processes, such as neurodegeneration, which, in turn, may sustain further PARP activation [49]. Therefore, for several years, PARP inhibitors have been considered as cytoprotective and/or anti-inflammatory agents. PARP inhibitors such as Olaparib and Veliparib proved to be neuroprotective in an in vitro model of cortical projection neurons death induced by different excitotoxic stimuli [50] and in an in vivo model of transient cerebral ischemia [51].

The study of this particular compound, even though we had already shown beneficial effects of PARP inhibition in HD, was aimed at the possible repurposing of a drug that is already available, and therefore to expedite the bench-to-bedside process.

Our data support the evidence that PARP inhibition can be used as a viable therapeutic strategy against neurodegenerative disorders such as HD and offers further evidence for repurposing and extending the use of PARP inhibitors to non-oncological disorders.

## Figures and Tables

**Figure 1 cells-09-02286-f001:**
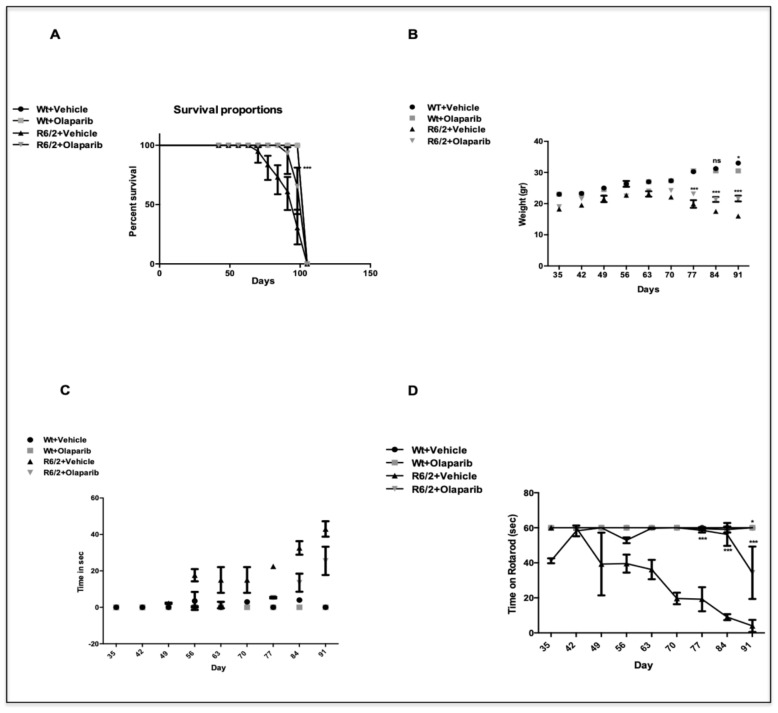
Secondary outcome measures. In (**A**), a Kaplan–Meier curve of survival is represented. R6/2 mice treated with Olaparib showed a mean survival time that was significantly longer (*p* < 0.0001) than that of R6/2 mice treated with vehicle. (**B**) Olaparib effects on wild type and R6/2 mice weight. A three-way ANOVA shows a statistically significant effect of Genotype (F_1,32_ = 45.8; *p* < 0.0001), Time (F_8,250_ = 86.24 *p* < 0.0001), and Genotype X Treatment X Time interaction (F_8,250_ = 12.78 *p* < 0.000) in R6/2 mice treated with Olaparib (**C**). Hind-limb clasping phenotype in R6/2 mice treated with Olaparib. Data analysis indicates a statistically significant effect of treatment and time. Olaparib-treated R6/2 mice showed significantly less clasping at weeks 11,12, and 13 (*** *p* < 0.001). (**D**) A three-way ANOVA with genotype, treatment, and time as main factors, revealed that R6/2 mice had a statistically significant impairment in motor coordination with respect to wild-type mice (F_1,36_ = 59.39 *p* < 0.0001) and that Olaparib treatment improves performance in a genotype-dependent fashion (F_8,288_ = 6.03; *p* < 0.0001).

**Figure 2 cells-09-02286-f002:**
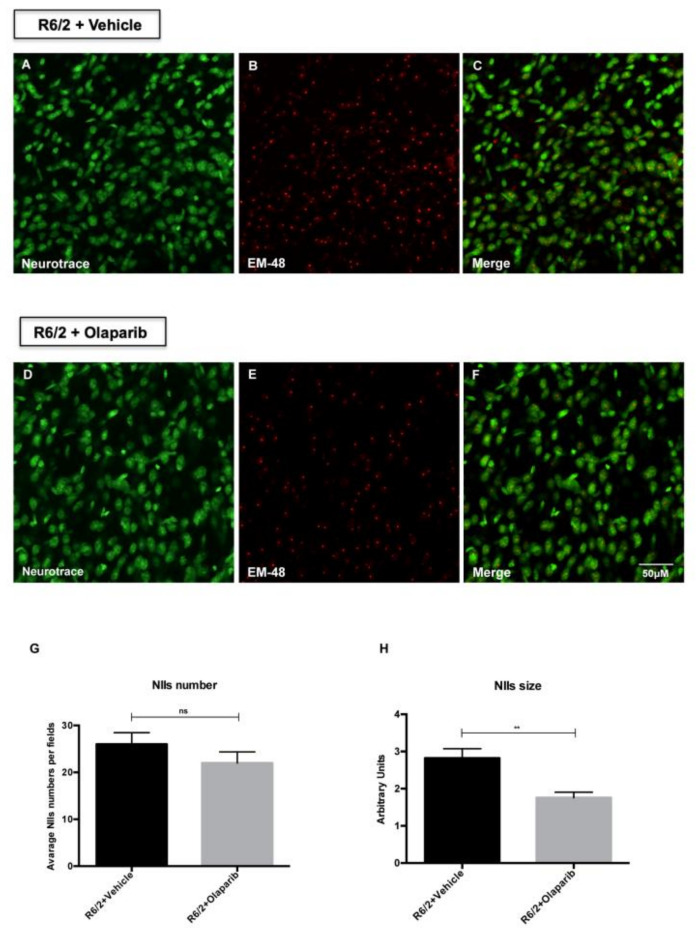
Primary outcome measures: Olaparib reduces neuronal intranuclear inclusions (NIIs) size in R6/2 mice. (**A**–**F**) Representative images of NIIs counterstained with green fluorescent Nissl in vehicle or Olaparib-treated R6/2 mice. (**G**,**H**) Two-way ANOVA analysis performed on data obtained by vehicle and Olaparib-treated R6/2 mice revealed a statistically significant effect of treatment on NIIs size. Unpaired t test with Welch′s correction performed on comparison analysis revealed a statistically significant decrease of NIIs size in the R6/2 mice group treated with Olaparib compared to that of vehicle-treated R6/2 mice (*p* < 0.01).

**Figure 3 cells-09-02286-f003:**
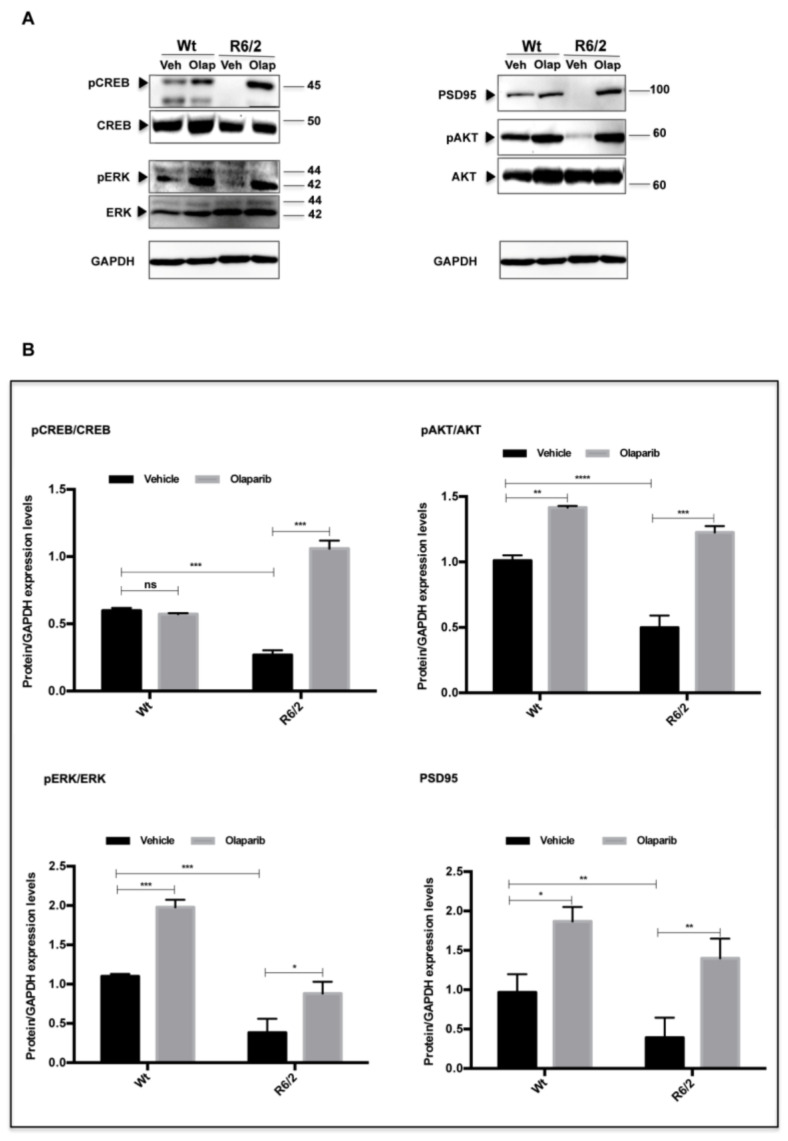
Olaparib promotes neuroprotection in R6/2 mice. (**A**) Brain tissue homogenates of all experimental groups were subjected to Western blotting analysis using pCREB, pERK, pAKT, and PSD-95. GAPDH was used as the loading control. Blots are representative of ≥3 independent experiments performed on 8 animals per group. (**B**) Graphs representing each protein expression level show that Olaparib treatment restored the phosphorylation of CREB (F_1,68_ = 4.791 *p* < 0.001), ERK (F_1,68_ = 29.77 *p* < 0.001), and AKT (F_1,68_ = 39.27 *p* < 0.001), promoting striatal neurons survival. Moreover, Olaparib was able to prevent post-synaptic modification in the R6/2 mice compared to that of the vehicle-treated group, which was dramatically lacking in the expression of PSD95 protein (F_1,68_ = 17.03; *p* < 0.01).

**Figure 4 cells-09-02286-f004:**
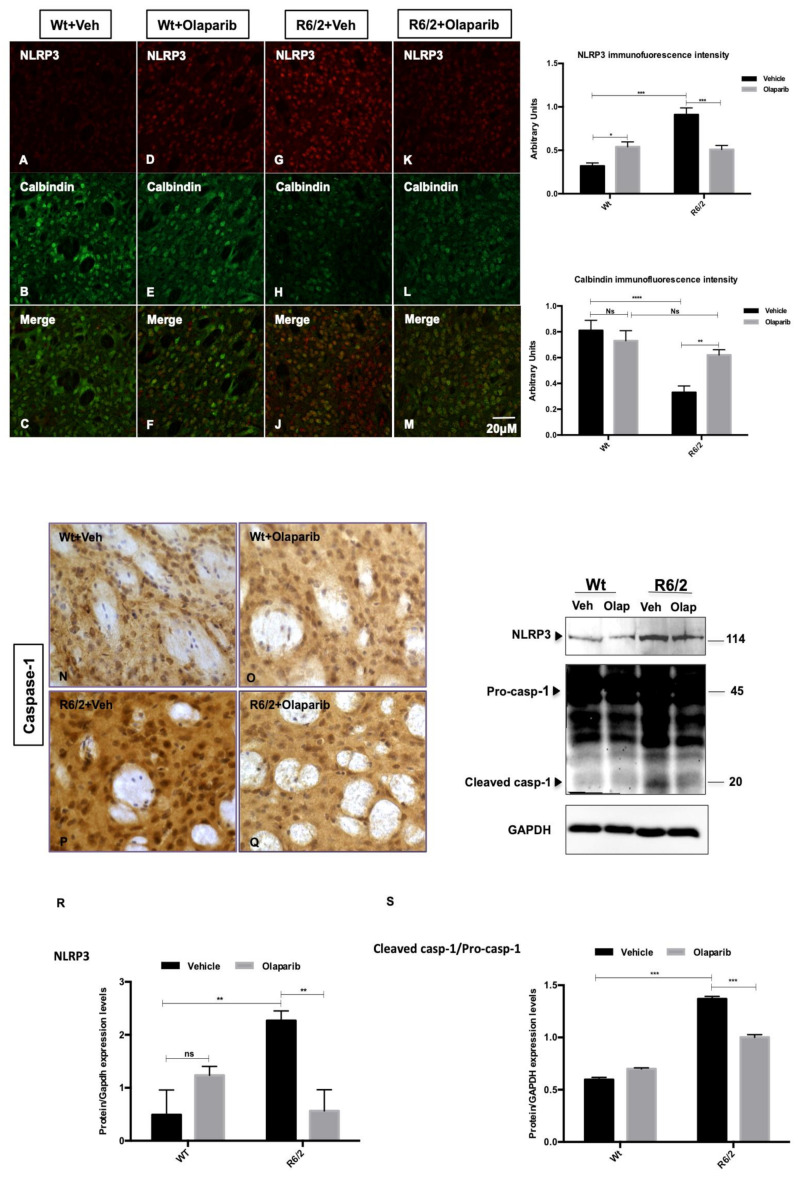
Effects of Olaparib on the expression/activation of inflammasome-mediated caspase-1 in R6/2 mice. (**A**–**M**) Collected images show the immunoreaction intensity relative to NLRP3 in each animal group at the different time points. Two-Way ANOVA analysis performed on data obtained by all animal groups showed a statistically significant effect of Genotype (F_1,30_ = 32.56; *p* < 0.001), Treatment (F_1,30_=26.58; *p* < 0.001), and Genotype X Treatment Interaction (F_1,30_ = 2.746; *p* < 0.001) of NLRP3 positive striatal neurons. Histogram shows NLRP3 intensity quantification, which is significantly decreased in the striatal neurons of Olaparib-treated R6/2 mice (treatment effect F_1,68_ = 5.13; *p* < 0.01) compared to vehicle group. (**N**–**Q**) Representative transmitted light microscope images showing Dab staining for caspase-1 contour stained with hematoxilin in the striatum of vehicle or Olaparib-treated Wt and R6/2 mice. (**R**,**S**) Two-way ANOVA performed on densitometric analysis of the cleavage product of caspase-1 revealed a statistically significant effect of Genotype (F1,68 = 1129.9; *p* < 0.001), Treatment (F_1,68_ = 6810; *p* < 0.001), and Genotype X Treatment Interaction (F1,68 = 41.34; *p* < 0.001) in the R6/2 group treated with Olaparib compared to the vehicle group.

**Figure 5 cells-09-02286-f005:**
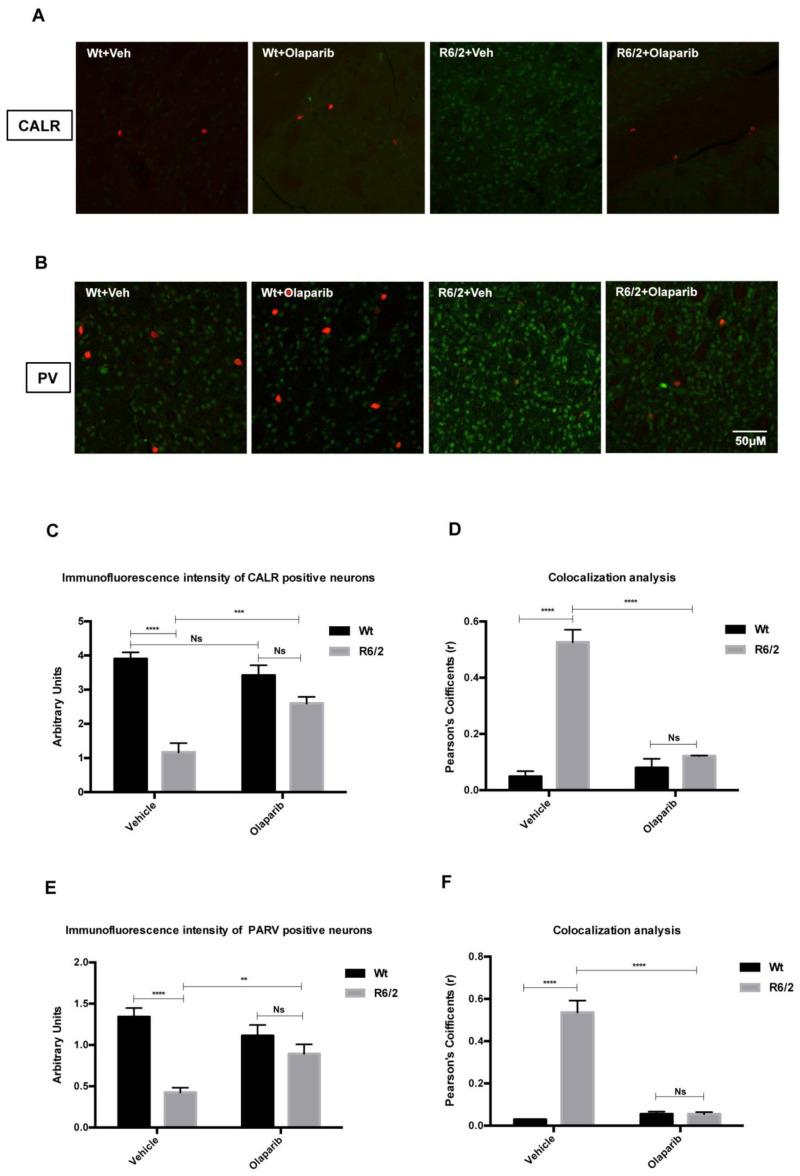
Olaparib prevents Parvalbuminergic and Calretininergic neurons loss. (**A**,**B**) Bonferroni analysis showed the significant dramatical reduction of number and immunofluorescence intensity of Calretinin and Parvalbumin containing neurons in the striatum of 13-week-old R6/2 mice compared to that of vehicle or Olaparib-treated Wt mice (**C**–**E**). (**F**,**G**) Graph shows no colocalization of NLRP3 in PARV and CALR interneurons of animal groups treated with Olaparib (mean Pearson’s coefficients equal to zero) and the statistically significant colocalization of NLRP3 in Parvalbumin and Calretinin containing interneurons (*p* < 0,001) of vehicle R6/2 mice.

**Figure 6 cells-09-02286-f006:**
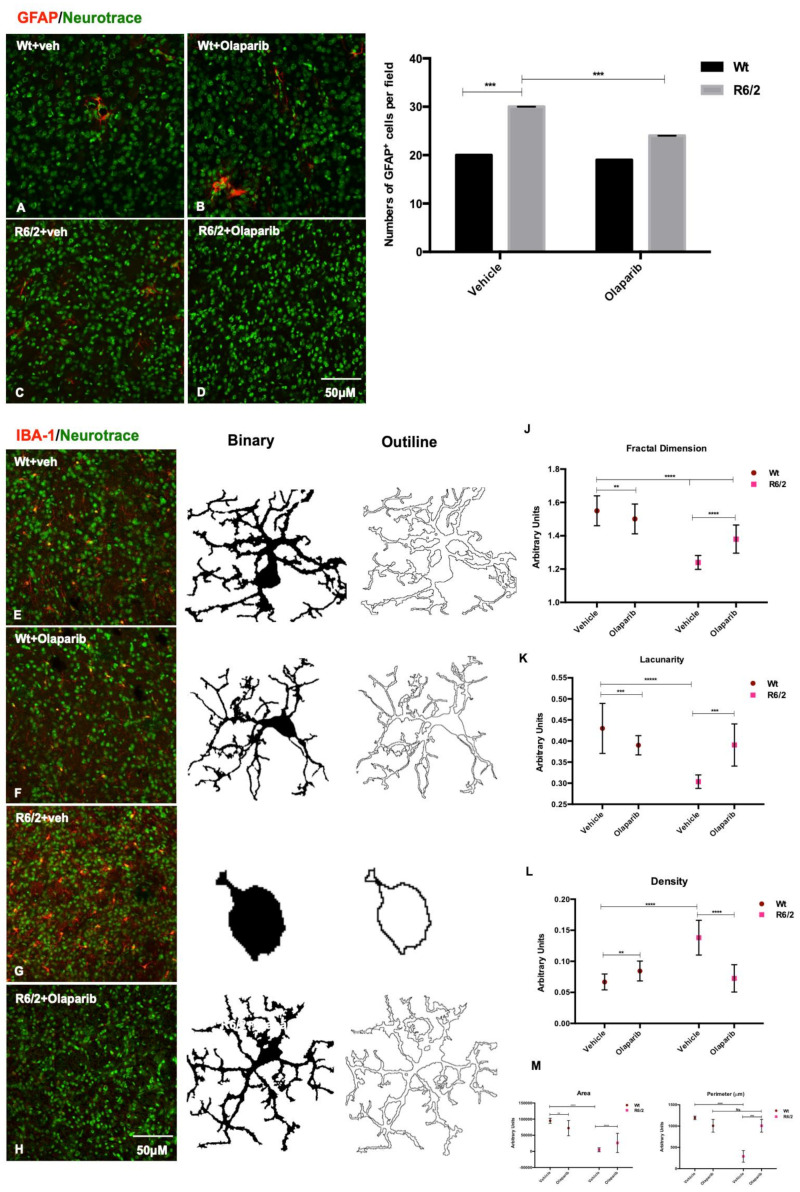
Olaparib reduces astrogliosis and microglial activation. (**A**–**D**) Representative confocal images showing the distribution of GFAP in the four experimental groups. Significantly lower GFAP positive cells were recorded in the Olaparib-treated R6/2 mice with respect to that of the saline-treated mice (F_1,76_ = 68008 *p* < 0.001). (**E**–**H**) Analyzed collected data revealed a significant reduction of microglia positive cells in the R6/2 mice treated with Olaparib compared to that in the saline-treated R6/2 mice. (**J**–**M**) The morphological parameters of fractal dimension, lacunarity, density, cell area, and perimeter revealed the statistically significant reduced microglial activation in the Olaparib-treated R6/2 mice. A reduced, activated microglia was evidenced by higher fractal dimension and lacunarity, indicating the increase of branch complexity and heterogeneity, and lower density, which implies a more compact shape and lower cell area and perimeter.

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
