# Peer review of "Modulation of Inflammasome and Pyroptosis by Olaparib, a PARP-1 Inhibitor, in the R6/2 Mouse Model of Huntington’s Disease"

_cells, 2020, doi:10.3390/cells9102286_

Round 1

Reviewer 1 Report

This report by Paldino et al. may be interesting to the readers of Cells, however, the study has some flaws that requires clarification.

In the first place, determination of disease stage of the animals seems problematic. Hind-limb clasping is the only legitimate measure of disease progression? And how was the stage actually determined? How much time does a presymptomatic or a severe mouse spend with clasping?

My most major concern is that the authors claim PARP inhibition in the brain of mice, but never actually show that inhibition indeed took place. How the dosage of Olaparib was set then? Did you verify the inhibition?

Besides, PARP inhibition has systemic effects, which might interfere with the study purposes. For example, PARP inhibition affects adipose tissue function and whole-body energy expenditure that may affect body weight. Adipose tissue weight, PPARg, SIRT1 activity should be measured in adipose tissue of mice. Moreover, PARP inhibition induced SIRT1 activation may also affect survival of the animals, that should be monitored as well.

My other major concern is that the authors report about pyroptosis upon PARP inhibition, but did not exclude the role of other types of cell death, such as apoptosis or parthanatos, in which PARP definitely has a role. Mitochondrial activity, caspase 3 and caspase 8 activity should be monitored to decidedly state that it is pyroptosis that underlies the phenotype.   

Author Response

This report by Paldino et al. may be interesting to the readers of Cells, however, the study has some flaws that requires clarification.

  • In the first place, determination of disease stage of the animals seems problematic. Hind-limb clasping is the only legitimate measure of disease progression? And how was the stage actually determined? How much time does a presymptomatic or a severe mouse spend with clasping?

Clasping phenoptype is a prominent sign of disease progression, the other one being rotarod performace. We measure the stage of disease by divinding in pre-syptomatic, symptomatic and advance stage according to the weeks of age (4 to 7 weeks, 8 to10 weeks fully symptomatic, 11 to death advanced stage). The presymptomatic mouse does not clasp at all, while the severe (advanced stage) can spend up to 60 seconds clasping

  • My most major concern is that the authors claim PARP inhibition in the brain of mice, but never actually show that inhibition indeed took place. How the dosage of Olaparib was set then? Did you verify the inhibition?

The literature has shown PARP inhibition by Olaparib in several reports,( Tajuddin et al, 2018; Dharwal and NAura, 2018, Patel et al, 2020), thus we are confident that PARP1 is inhibited by the compound. The dosage of Olaparib was chosen also by a review of the literature (Amal et al, 2019)

Tajuddin N, Kim HY, Collins MA. PARP Inhibition Prevents Ethanol-Induced Neuroinflammatory Signaling and Neurodegeneration in Rat Adult-Age Brain Slice Cultures. J Pharmacol Exp Ther. 2018;365(1):117-126. doi:10.1124/jpet.117.245290

Dharwal V, Naura AS. PARP-1 inhibition ameliorates elastase induced lung inflammation and emphysema in mice. Biochem Pharmacol. 2018;150:24-34

Patel PR, Senyuk V, Sweiss K, et al. PARP Inhibition Synergizes with Melphalan but Does not Reverse Resistance Completely. Biol Blood Marrow Transplant. 2020;26(7):1273-1279. doi:10.1016/j.bbmt.2020.03.008

Ahmad A, Vieira JC, de Mello AH, et al. The PARP inhibitor olaparib exerts beneficial effects in mice subjected to cecal ligature and puncture and in cells subjected to oxidative stress without impairing DNA integrity: A potential opportunity for repurposing a clinically used oncological drug for the experimental therapy of sepsis. Pharmacol Res. 2019;145:104263.

  • Besides, PARP inhibition has systemic effects, which might interfere with the study purposes. For example, PARP inhibition affects adipose tissue function and whole-body energy expenditure that may affect body weight. Adipose tissue weight, PPARg, SIRT1 activity should be measured in adipose tissue of mice. Moreover, PARP inhibition induced SIRT1 activation may also affect survival of the animals, that should be monitored as well.

This is indeed a very good point, as pleotropic effects of compound always have to be taken into account. Moreover, HD patients undergo a severe sarcopenia, as well as HD models. This is something that we shall include in the next series of experiments.

  • My other major concern is that the authors report about pyroptosis upon PARP inhibition, but did not exclude the role of other types of cell death, such as apoptosis or parthanatos, in which PARP definitely has a role. Mitochondrial activity, caspase 3 and caspase 8 activity should be monitored to decidedly state that it is pyroptosis that underlies the phenotype.

This is also a very good point. We did monitor Caspase 8 activation   and it is now in the supplemental material of our manuscript. We already had demonstrated that Caspase 8 activation was downmodulated in R6/2 at 13 weeks, in favor of an upregulation of Caspase 1 (Paldino et al, 2020), which corroborates the concept of a higher expression of pyroptosis in the late stages of disease.

Paldino E, D'Angelo V, Sancesario G, Fusco FR. Pyroptotic cell death in the R6/2 mouse model of Huntington's disease: new insight on the inflammasome. Cell Death Discov. 2020;6:69. Published 2020 Jul 31. doi:10.1038/s41420-020-00293-z

Reviewer 2 Report

This study reports the neuroprotective effect of Olaparib, an FDA-approved drug used for ovian cancer, in  a mouse model of Huntington's disease. Administration of Olaparib increased survival, attenuated neurological deficits and improved clinical outcomes through activating the inflammasome pathway. The experiments are well-designed, the manuscript is well-written, and the results showed that this drug could serve as useful agent for Huntington's disease therapy. 

Author Response

This study reports the neuroprotective effect of Olaparib, an FDA-approved drug used for ovian cancer, in  a mouse model of Huntington's disease. Administration of Olaparib increased survival, attenuated neurological deficits and improved clinical outcomes through activating the inflammasome pathway. The experiments are well-designed, the manuscript is well-written, and the results showed that this drug could serve as useful agent for Huntington's disease therapy. 

We want to thank the reviewer for the kind comments about our manuscript

Reviewer 3 Report

In this manuscript, the authors investigated the neuroprotective effect of PARP inhibitor, olaparib on R6/2 HD mice. They found that olaparib improved survival, neurological deficit and behaviour by regulating the inflammasome activation.

Major comments:

  1. Lack of novelty: The effects of PARP inhibition on survival, weights, neurological deficit and microglial reaction in R6/2 HD mice have already been published (Cardinale et al, Plos One, 2015). Thus, the novelty of this manuscript is limited.
  2. This manuscript is poorly written. The introduction section contains too much unnecessary information, and does not describe the research question in their study. The discussion section repeats the introduction and results without interpretation, and there is no flow.

Author Response

  1. Lack of novelty: The effects of PARP inhibition on survival, weights, neurological deficit and microglial reaction in R6/2 HD mice have already been published (Cardinale et al, Plos One, 2015). Thus, the novelty of this manuscript is limited.

The novelty of our study consists in two main aspects, one that has a clinical implication, and the other more  about the mechanisms. Indeed, we discovered that this particular compound is effective in HD, and this is very important because it is already commercially available and this could speed up possible clinical trials. HD patients are in need of new drugs as, so far, only tetrabenazine is available as a recognized HD medication.

The other aspect involves the molecular mechanisms underlying neurodegeneration. Indeed, we have confirmed that PARP1 inhibition is able to modulate pyroptosis and NLRP3 inflammasome in our HD model.

2.  This manuscript is poorly written. The introduction section contains too much unnecessary information, and does not describe the research question in their study. The discussion section repeats the introduction and results without interpretation, and there is no flow.

We have edited the manuscript in order to make it clearer particularly in terms of question of the study (in the introduction) and interpretation (in the discussion)

Round 2

Reviewer 1 Report

In the manuscript by Paldino et al. the authors demonstrate that PARP inhibition by Olaparib in a mouse model of Huntington's disease exerted beneficial effects.

However, the data are still not convincing enough to state that the phenotype of Olaparib-treated R6/2 mice is a result of  the down-redulation of pyroptosis upon PARP inhibition in the brain.

First, there is no proof for PARP inhibition in brain neurons. I understand what literature says, but it is necessary to show what you state. Second, it would also be necessary to characterize pyroptosis in the neurons, for example, by morphological studies to exclude the role of apoptosis in this model. Caspase-1 activation is not sufficient.

I suggest the authors to perform in vitro studies with PARP inhibitors (more than one) on a neural cell line with modification in huntingtin for example, and extensively study the markers of inflammation, cell death, morphology etc.

Author Response

(...) However, the data are still not convincing enough to state that the phenotype of Olaparib-treated R6/2 mice is a result of  the down-redulation of pyroptosis upon PARP inhibition in the brain.

In this last version, we provide proof of PARP -1 inhibition by olaparib in a western blot experiment (please see suppl. figure 1). However, we do agree that we cannot state that PARP-1 inhibition per se is able to have the effects we observed, so we changed the title of the paper accordingly.

First, there is no proof for PARP inhibition in brain neurons. I understand what literature says, but it is necessary to show what you state. Second, it would also be necessary to characterize pyroptosis in the neurons, for example, by morphological studies to exclude the role of apoptosis in this model. Caspase-1 activation is not sufficient.

In a previous paper we have shown a morphological study of pyroptosis in the R6/2 HD model (Paldino et al, 2020)

I suggest the authors to perform in vitro studies with PARP inhibitors (more than one) on a neural cell line with modification in huntingtin for example, and extensively study the markers of inflammation, cell death, morphology etc.

The studies suggested are certainly very interesting and potentially will be able to complete the current knowledge acquired in our in vivo studies.

Reviewer 3 Report

The authors have made efforts to make the manuscript more interesting and easily understandable. The novelty of the study is also clearer in the revised version. Therefore, I agree to accept in present form. 

Author Response

We wish to thank the reviewer very much for the comments